# Ten Plastomes of *Crassula* (Crassulaceae) and Phylogenetic Implications

**DOI:** 10.3390/biology11121779

**Published:** 2022-12-07

**Authors:** Hengwu Ding, Shiyun Han, Yuanxin Ye, De Bi, Sijia Zhang, Ran Yi, Jinming Gao, Jianke Yang, Longhua Wu, Xianzhao Kan

**Affiliations:** 1Anhui Provincial Key Laboratory of the Conservation and Exploitation of Biological Resources, College of Life Sciences, Anhui Normal University, Wuhu 241000, China; 2College of Landscape Engineering, Suzhou Polytechnic Institute of Agriculture, Suzhou 215000, China; 3CAS Key Laboratory of Soil Environment and Pollution Remediation, Institute of Soil Science, Chinese Academy of Sciences, Nanjing 210008, China; 4The Institute of Bioinformatics, College of Life Sciences, Anhui Normal University, Wuhu 241000, China

**Keywords:** *Crassula*, Crassulaceae, plastome, codon usage, codon aversion, DNA barcoding, evolutionary rates, phylogeny

## Abstract

**Simple Summary:**

Plastids are semi-autonomous plant organelles which play critical roles in photosynthesis, stress response, and storage. The plastid genomes (plastomes) in angiosperms are relatively conserved in quadripartite structure, but variable in size, gene content, and evolutionary rates of genes. The genus *Crassula* L. is the second-largest genus in the family Crassulaceae J.St.-Hil, that significantly contributes to the diversity of Crassulaceae. However, few studies have focused on the evolution of plastomes within *Crassula*. In the present study, we sequenced ten plastomes of *Crassula*: *C. alstonii* Marloth, *C. columella* Marloth & Schönland, *C. dejecta* Jacq., *C. deltoidei* Thunb., *C. expansa* subsp. *fragilis* (Baker) Toelken, *C. mesembrianthemopsis* Dinter, *C. mesembryanthoides* (Haw.) D.Dietr., *C. socialis* Schönland, *C. tecta* Thunb., and *C. volkensii* Engl. Through comparative studies, we found *Crassula* plastomes have unique codon usage and aversion patterns within Crassulaceae. In addition, genomic features, evolutionary rates, and phylogenetic implications were analyzed using plastome data. Our findings will not only reveal new insights into the plastome evolution of Crassulaceae, but also provide potential molecular markers for DNA barcoding.

**Abstract:**

The genus *Crassula* is the second-largest genus in the family Crassulaceae, with about 200 species. As an acknowledged super-barcode, plastomes have been extensively utilized for plant evolutionary studies. Here, we first report 10 new plastomes of *Crassula*. We further focused on the structural characterizations, codon usage, aversion patterns, and evolutionary rates of plastomes. The IR junction patterns—IRb had 110 bp expansion to *rps19*—were conservative among *Crassula* species. Interestingly, we found the codon usage patterns of *matK* gene in *Crassula* species are unique among Crassulaceae species with elevated ENC values. Furthermore, subgenus *Crassula* species have specific GC-biases in the *matK* gene. In addition, the codon aversion motifs from *matK*, *pafI*, and *rpl22* contained phylogenetic implications within *Crassula*. The evolutionary rates analyses indicated all plastid genes of Crassulaceae were under the purifying selection. Among plastid genes, *ycf1* and *ycf2* were the most rapidly evolving genes, whereas *psaC* was the most conserved gene. Additionally, our phylogenetic analyses strongly supported that *Crassula* is sister to all other Crassulaceae species. Our findings will be useful for further evolutionary studies within the *Crassula* and Crassulaceae.

## 1. Introduction

The family Crassulaceae comprises approximately 1400 species in 34 genera and three subfamilies (Crassuloideae Burnett, Kalanchoideae A. Berger, and Sempervivoideae Arn.) [1,2,3,4,5,6,7]. These subfamilies can be further subdivided into seven major clades: Crassula (Crassuloideae), Kalanchoe (Kalanchoideae), and the other five clades (Sempervivum, Leucosedum, Aeonium, Acre, and Telephium), which form the largest subfamily Sempervivoideae [3,4,5,6,8]. The genus *Crassula*, with about 200 accepted species, is the only unique genus in the clade Crassula, the second-largest genus of Crassulaceae, and significantly contributes to the diversity of Crassulaceae [3,9,10]. Previous taxonomic revision of *Crassula* recognized two subgenera: *Crassula* L. and *Disporocarpa* Fischer & C.A. Mey. [7,11,12]. The monophyly of the subgenus *Crassula* was well supported in two recent molecular phylogenetic studies [9,10]. Nevertheless, the monophyly of subgenus *Disporocarpa* is still controversial [9,10]. Thus, more evidence and further investigations are required to clarify the phylogenetic relationships of *Crassula*.

Plastids are semi-autonomous plant organelles which have many vital functions, such as photosynthesis, stress response, and storage [13]. In angiosperms, the plastid genome (plastome) generally exhibits a conserved quadripartite circular structure with a size of 120–170 kb, comprising two single copy regions (larger and small regions, namely LSC and SSC, respectively) and two inverted repeat regions (IRs) [14,15,16]. Owing to the low level of recombination, uniparental inheritance, and without interference from paralogs, plastome has been extensively utilized as a super-barcode for plant species identification and evolutionary studies [17,18,19,20,21,22,23,24]. Due to the rapid development and widespread application of high-throughput sequencing technologies (such as Illumina, PacBio, and Nanopore sequencing technologies), an increasing number of complete Crassulaceae plastomes (more than 70 sequences) have been deposited in public databases. However, within the *Crassula*, only one plastome has been reported to date [6]. The lack of plastome data has limited the progress in investigating the evolutionary history of *Crassula*. Therefore, more plastome data from *Crassula* are needed to address this issue.

Codon usage bias (CUB), indicating the preferential utilization of synonymous codons in protein-coding genes (PCGs), has evolved via combined effects of genetic drift, mutation, and natural selection [17,25,26,27,28]. Owing to different species having diverse codon usage patterns, investigations of CUB can reveal phylogenetic relationships between species [17,25,26,27,28]. Codon aversion is defined as the codon which is not used in a certain gene [29,30,31]. The codon aversion motif is phylogenetically conserved in some lineages [29,30,31]. Interestingly, our recent reports in Macaronesian species (Crassulaceae) and *Bletilla* Rchb.f. species (Orchidaceae Juss.) have suggested that plastid CUB and codon aversion patterns might harbor phylogenetic signals [17,26]. Therefore, the analyses of plastid genes in codon-usage aspects might broaden our understanding of the phylogeny of both *Crassula* and Crassulaceae.

Evolutionary rate, calculated by the ratio (dN/dS) of nonsynonymous rate (dN) and synonymous rate (dS), can quantify the intensity of the selective force acting on a PCG [32,33,34]. The evolutionary rate can also reflect the pattern of natural selection (dN/dS value >1, =1, and <1 indicate positive, neutral, and purifying selection, respectively) [33,34,35]. The dN/dS values in different genes are variable, which might be influenced by many factors, such as protein function, population size, generation time, and DNA-repair efficiency [36,37]. The dN/dS values of plastid genes have been measured in many plant lineages, and most values were lower than 1, indicating plastid genes were mainly under the purifying selection [13,22,38,39,40]. Currently, the detailed rates and patterns of plastid genes were largely unknow in Crassulaceae. Knowledge of the evolutionary rates and patterns will shed light on how the diversifying selection affected the plastome evolution in Crassulaceae.

To address these issues, we newly sequenced and assembled the plastomes of ten *Crassula* species (*C. alstonii*, *C. columella*, *C. dejecta*, *C. deltoidea*, *C. expansa* subsp. fragilis, *C. mesembrianthemopsis*, *C. mesembryanthoides*, *C. socialis*, *C. tecta* and *C. volkensii*) using Illumina sequencing technology. Together with the public data, we performed comprehensive analyses to investigate (1) structural characterizations of *Crassula* plastomes, (2) unique CUB and codon-aversion patterns for Crassula plastomes, (3) evolutionary rates and patterns of plastid genes of Crassulaceae, and (4) phylogenetic relationships among Crassulaceae species. Our findings will not only shed new insights into the plastome evolution of Crassulaceae, but also provide potential molecular markers for DNA barcoding.

## 2. Materials and Methods

### 2.1. Sample Collection, DNA Extraction, and Sequencing

The fresh leaf samples of ten *Crassula* species were collected from greenhouses of Anhui Normal University, with the voucher codes KL01739, KL01709, KL01449, KL01646, KL02048, KL01731, KL01444, KL01653, KL01657, and KL01688 for *C. alstonii*, *C. columella*, *C. dejecta*, *C. deltoidea*, *C. expansa* subsp. *fragilis*, *C. mesembrianthemopsis*, *C. mesembryanthoides*, *C. socialis*, *C. tecta*, and *C. volkensii*, respectively. The Plant Genomic DNA kit (Tiangen, Beijing, China) was used for Genomic DNA extraction. Furtherly, a TruSeq DNA PCR-Free Library Prep Kit (Illumina, San Diego, CA, USA) was employed for library construction. Then, these libraries were sequenced using the Illumina Hiseq X Ten (Illumina, San Diego, CA, USA) platform.

### 2.2. Plastome Assembly, Genome Annotation, and Comparative Genomic Analysis

All resulting high-quality clean reads were assembled by using GetOrganelle 1.7.5 [41] with the plastome of *C. perforata* Thunb. (NC_053949) [6] as reference. The plastomes were initially annotated with the online program GeSeq [42] and then checked manually. Bowtie 2.4.1 [43] and Chloroplot [44] were utilized for the sequencing depth estimation and the drawing of a gene map, respectively. Genome comparisons were visualized using mVISTA [45] in Shuffle-LAGAN mode. In order to detect highly variable regions (HVRs) among plastomes, the sliding-window nucleotide diversity (π) values were measured in DnaSP v6.12 (window length = 600 bp, and step size = 200 bp) [46]. The contiguous sliding windows with higher π values (π > π_mean_ + 2 standard deviation) were merged as a HVR [47,48]. The contraction and expansion of IR regions at the junctions of plastomes were subsequently plotted using R package IRscope V0.1.R (Viikki Plant Science Centre, University of Helsinki, Helsinki, Finland) [49].

### 2.3. Codon Usage and Aversion Indices Analyses

To investigate the codon usage indices, we used CodonW v.1.4.2 (Peden, University of Nottingham, Nottingham, UK) to calculate the values of relative synonymous codon usage (RSCU), and the effective number of codons (ENC) of plastid genes (length ≥300 bp) among 87 Crassulaceae species (10 of which are new in this study, Appendix A). The RSCU value for a codon represents the observed frequency divided by the expected frequency (RSCU >1 implies a codon use higher than expected, and vice versa) [50]. The RSCU heatmap was rendered using TBtools 1.098 [51]. In addition, the ENC values, ranging from 20 (extreme bias) to 61 (no bias), quantify the level of CUB of synonymous codons [52]. Furtherly, the parity rule 2 (PR2) plot was performed according to the two formulas: GC-bias = [G3/(G3 + C3)|4] and AT-bias = [A3/(A3 + T3)|4] (“|4” means 4-fold degenerate synonymous codons, and G3, C3, A3 and T3 denotes nucleotide composition at the 3rd codon sites, respectively) [53,54]. The points lying at the centre of plot (AT bias = 0.5 and GC bias = 0.5) indicate no bias, whereas the off-centred points reflect the direction and extent of bias [53,54]. Moreover, the codon aversion motifs harboring strong phylogenetic implications were identified by using CAM v.1.02 [31].

### 2.4. Nucleotide Substitution Rate Analyses

The 79 PCGs from 87 species of Crassulaceae were employed to evaluate the evolutionary rates (Appendix A). The percentage of variable sites (PV) and average π values were measured with DnaSP v6.12 (Departament de Genètica, Universitat de Barcelona, Barcelona, Spain) [46]. The nucleotide substitution rates, including dN, dS, and dN/dS, were inferred with PAML v4.9 [55] under F3X4 and M0 model.

### 2.5. Phylogenetic Implications Analyses

Phylogenetic relationships among 87 Crassulaceae species were inferred by maximum-likelihood (ML) and Bayesian inference (BI) methods, based on 79 PCGs (Data S1). Recent studies of Lu et al. [9] and Bruyns et al. [10] revealed a sister relationship between Crassulaceae and Haloragaceae R.Br. Therefore, two species of Haloragaceae (*Myriophyllum aquaticum* (Vell.) Verdc., NC_048889 and *Myriophyllum spicatum* L., NC_037885) were selected as outgroups. Multiple sequence alignments were generated using MAFFT v7.505 in PhyloSuite v1.2.1 with codon model [56]. The best-fit nucleotide substitution models were evaluated with ModelTest-NG v0.1.7 [57]. Subsequently, we employed RAxML-NG 1.1 [58] and MrBayes v3.2.7a [59] for ML and BI analyses, respectively. For ML analyses, the reliabilities were assessed with 1000 bootstrap replicates, and the convergence was evaluated by using parameter “--bsconverge” in RAxML-NG package (Computational Molecular Evolution Group, Heidelberg Institute for Theoretical Studies, Heidelberg, Germany). For BI analyses, four independent Markov chains and two independent runs (running for 10,000,000 generations, and sampling every 1000 generations) were conducted, with Tracer 1.7.1 (Institute of Evolutionary Biology, University of Edinburgh, Edinburgh, UK) [60] for the convergence. After discarding the first 25% trees as burn-in, the remaining 75% trees were used to estimate the consensus tree and Bayesian posterior probabilities.

## 3. Results

### 3.1. Plastome Organizations and Structural Features

Based on bowtie2 mapping, totally 3,246,461, 1,740,232, 3,915,950, 2,632,260, 2,801,895, 504,972, 5,440,628, 3,398,113, 1,877,288 and 1,530,319 paired reads were mapped to the plastomes of *C. alstonii* (coverage: 3344.02×), *C. columella* (coverage: 1762.18×), *C. dejecta* (coverage: 4020.78×), *C. deltoidei* (coverage: 5284.33×), *C. expansa* subsp. *fragilis* (coverage: 5796.17×), *C. mesembrianthemopsis* (coverage: 1010.16×), *C. mesembryanthoides* (coverage: 5557.15×), *C. socialis* (coverage: 3486.98×), *C. tecta* (coverage: 1918.07×), and *C. volkensii* (coverage: 3161.69×), respectively. The new complete plastomes of ten species of *Crassula* (accession numbers: OP729482–OP729487 and OP882297–OP882300) were typical circular and quadripartite biomolecules (Figure 1), with sizes ranging from 144,855 bp to 146,060 bp. These plastomes contains LSC (78,303–79,707 bp), SSC (16,568–16,871 bp), and IR (24,810–24,878 bp). The overall GC contents of *Crassula* plastomes were between 37.73% and 38.32%. Notably, the GC contents of IR regions (42.93–43.15%) were found to be higher than those of in the LSC (35.75–36.51%) and SSC regions (31.67–32.40%). In addition, these plastomes contain 134 genes, including 85 PCGs, 37 tRNA genes, 8 rRNA genes and 4 pseudogenes. Among these genes, 6 PCGs, 7 tRNA genes, 4 rRNA genes, and one pseudogene (*ycf15*), were completely duplicated in the IR regions (Table 1).

Furthermore, based on the results obtained with mVISTA, in all plastomes investigated it was found that the IR and coding regions (exons, tRNAs, and rRNAs) are more conserved than SC and conserved non-coding regions (CNS), respectively (Figure 2). Additionally, the results also revealed that 3 plastomes (labelled 8–10) of subgenus *Disporocarpa* exhibited higher divergences than 7 plastomes (labelled 1–7) of subgenus *Crassula*, when compared with the reference.

The sliding-window-based π values estimated for 11 plastomes of *Crassula* ranged from 0.00073 to 0.10315 (Appendix A). The mean π value and its standard deviation were 0.02978 and 0.01954, respectively. Thus, a total of 11 HVRs were identified with relatively high variability (π > 0.06886) (Figure 3). These HVRs containing high π values (0.06912–0.08653) and abundant variable sites (111–559) might be used as potential DNA barcodes for species identification within *Crassula* (Table 2).

In our current study, all 11 plastomes of *Crassula* displayed similar IR junction patterns (Figure 4). The SSC/IRa borders are located in the coding regions of *ycf1* gene, resulting in the fragmentations of *ycf1* (*ycf1*-fragment) in IRb regions. Moreover, *ndhF* genes were discovered to occur mainly in SSC, and partly in IRb, regions. Notably, *rps19* genes are located at the LSC/IRb junctions, with extension into the IRb regions for 110 bp. Similarly, *trnH* genes lie at the IRa/LSC junctions, with uniform 3 bp-sized expansions to the IRa regions.

### 3.2. Codon Usage and Aversion Patterns

To compare the patterns of codon usage and aversion between *Crassula* and other Crassulaceae species, four analyses (RSCU, ENC, PR2-plot, and codon aversion motif) of 53 plastid genes (length ≥300 bp) were performed.

The overall RSCU values ranged from 0.32 (CTC or AGC) to 2.07 (TTA) among Crassulaceae species (Appendix A). Similar with other Crassulaceae species, seven taxa of *Crassula* exhibited significant preference for A/T-ending codons over G/C-ending codons in plastid genes (Figure 5). Importantly, the RSCU heatmap showed two subgenera within the *Crassula*: subgenus *Disporocarpa* included *C. expansa* subsp. *fragilis*, *C. deltoidea* and *C. volkensii*; subgenus *Crassula* consisted of the remaining eight taxa (Figure 5).

The ENC values ranged from 30.83 (*ndhC* in *Sedum sarmentosum* Bunge) to 57.74 (*ndhJ* in *C. volkensii* and *C. expansa* subsp. *fragilis*) among Crassulaceae species (Appendix A). Generally, ENC values ≤35 indicate high codon preference [52,61,62]. The results show that most of the ENC values (99.48%) were higher than 35, indicating a weaker bias. Most surprisingly of all, we detected the ENC values of *matK*, from the Crassula clade, are significantly higher than those of all other clades (Appendix A and Figure 6). It might prove to be a unique feature for *Crassula* species. To further verify this finding, more sampling data and comprehensive analyses are need in future studies.

The PR2 plots of *matK* and 52 other PCGs are presented in Figure 7 and Appendix A, respectively. These results indicated the nucleotide usage at the 3rd codon site of 4-fold degenerate codons is uneven in different genes. For example, *rps14*, *clpP*, *psbA*, and *pafII* prefer to use A/G, A/C, T/C, and T/G in 4-fold degenerate sites, respectively (Appendix A). In addition, these unbalanced utilizations were also found in different species (Appendix A). Obviously divergent GC-biases were observed in *matK* genes between species of subgenus *Crassula* and others. Specifically, all GC-biases of clades from Kalanchoideae and Sempervivoideae, plus subgenus *Disporocarpa*, were less than 0.5. On the contrary, all these values for the subgenus *Crassula* were higher than 0.5, which might be unique characteristic for subgenus *Crassula*. Moreover, species with close relationships had identical nucleotide biases. For example, *C. alstonii* and *C. columella* had identical AT-biases (0.4074) and GC-biases (0.5455). Similar phenomena could also be observed in *C. mesembryanthoides* and *C. tecta* (AT-biases = 0.4286, and GC-biases = 0.5455).

Owing to the codon aversion motifs containing phylogenetic implication, we analyzed codon aversion patterns of genes among Crassulaceae species. Except for *rpoB*, *rpoC2*, *ycf1* and *ycf2*, codon aversion motifs were found in the remaining 49 genes (Appendix A). It is worth noting that 27 and 16 unique codon aversion motifs were detected for species of subgenus *Crassula* and subgenus *Disporocarpa*, respectively (Table 3), which might be used as potential biomarkers for species identification. Further to this, 8 consensus motifs might be considered as the feature of genus *Crassula* (Table 3). Moreover, the codon aversion motifs from 3 genes (*matK*, *pafI* and *rpl22*) could also divide 11 species into two subgenera (subgenus *Crassula* and subgenus *Disporocarpa*) (Figure 8), which is congruent with results from RSCU heatmap.

### 3.3. Evolutionary Rates and Patterns 

The π (0.00447–0.0914) and PV (4.91–37.52%) values of 79 plastid PCGs of Crassulaceae species were plotted in Figure 9a. Two genes, referring to *ycf1* (π = 0.0914, PV = 35.78%) and *matK* (π = 0.08239, PV = 37.52%), had obviously higher π and PV values than those of the other 77 genes, indicating they might accumulate more mutations than other plastid genes. The detailed data are listed in Appendix A.

To further quantify the evolutionary rates of PCGs, the nucleotide substitution rates, including dN, dS and dN/dS, were calculated (Figure 9b, Appendix A). The dN values ranged from 0 to 0.8671, with higher dN values for *ycf1* (dN = 0.8671) and *matK* (dN = 0.7804) than for others. Compared with dN values, the dS values had relatively wide ranges (0.177–2.3917), resulting in corresponding dN/dS ratios (0–0.5891) of less than 1. This finding indicates the plastid genes from Crassulaceae appear to be evolving under a purifying selective constraint. Among 79 plastid PCGs, *ycf2* is the most rapidly evolving gene, with the highest ratio (dN/dS = 0.5891), followed by *ycf1*, *cemA*, *psaI*, and *matK*. By contrast, *psaC* was the most conserved gene with the lowest ratio (dN/dS = 0).

### 3.4. Phylogenetic Implications

To investigate the evolutionary relationships among 87 species of Crassulaceae, phylogenetic analyses were performed. After a model test, GTR + G4 and GTR + I+G4 were inferred as the optimal substitution models for most genes (the detailed models can be seen in Appendix A). As shown in Figure 10, the trees inferred from two methods displayed the same topology.

Ten species of *Crassula* that we sequenced, together with *C. perforate*, form the well-supported clade Crassula, which is sister to all other Crassulaceae species (maximum likelihood bootstrap [BS] = 100 and bayesian posterior probability [PP] = 1.00). In addition, our phylogenetic tree indicated that this monophyletic clade could be clustered into two subgenera: subgenus *Disporocarpa* harbored *C. volkensii*, *C. expansa* subsp. *fragilis* and *C. deltoidea* ([BS] = 100 and [PP] = 1.00). Subgenus *Crassula* included the remaining eight *Crassula* species (*C. alstonii*, *C. columella*, *C. dejecta*, *C. mesembryanthoides*, *C. tecta*, *C. mesembrianthemopsis*, *C. socialis*, and *C. perforata*) ([BS] = 100 and [PP] = 1.00). Within subgenus *Crassula*, two species (*C. alstonii* and *C. columella*) were sister to six other species (*C. dejecta*, *C. mesembryanthoides*, *C. tecta*, *C. mesembrianthemopsis*, *C. socialis*, and *C. perforata*) ([BS] = 100 and [PP] = 1.00). Further, *C. tecta* and *C. mesembrianthemopsis* formed the well-supported sister taxa ([BS] = 100 and [PP] = 1.00). Unfortunately, the sister group of *C. dejecta* and *C. mesembryanthoides* had relatively weak support ([BS] = 55 and [PP] =0.69). Due to the limited plastome data within *Crassula*, there are many unsolved phylogenetic problems in this clade. Therefore, more samples are needed to solve this issue.

As expected, the six species from genus *Kalanchoe* Adans. and genus *Cotyledon* L. formed the monophyletic clade Kalanchoe (or subfamily Kalanchoideae) ([BS] = 100 and [PP] = 1.00). The remaining 70 species, belonging to the subfamily Sempervivoideae, can be further grouped into 5 distinct clades: Acre, Aeonium, Leucosedum, Sempervivum, and Telephium. In detail, 7 *Sedum* L. species and 3 species from other genera respectively (*Graptopetalum* Rose, *Echeveria* DC., and *Pachyphytum* Link, Klotzsch & Otto) formed a well-supported clade Acre ([BS] = 100 and [PP] = 1.0). However, it is clear from our results that *Sedum* is not monophyletic, with some other taxa embedded within this genus.

In addition, 13 species from genus *Aeonium* Webb & Berthel. and genus *Monanthes* Haw. make up the clade Aeonium ([BS] = 100 and [PP] = 1.0). Furthermore, due to sampling in this study, only a single species form Leucosedum and Sempervivum clades, and full resolution of relationships within these clades requires sufficient molecular sequences. Notably, the clade Telephium, with 45 species, consists of clusters “*Rhodiola*” and “*Hylotelephium*” [63] ([BS] = 92 and [PP] = 1.0). Within cluster “*Hylotelephium*“, non-monophyly of *Orostachys* Fisch. was observed. Three *Orostachys* species, (*O. japonica* (Maxim.) A.Berger, *O. minuta* (Kom.) A.Berger, and *O. fimbriata* (Turcz.) A.Berger) belonging to the Subsection *Orostachys* [63] ([BS] = 100 and [PP] = 1.0), were sister to *Meterostachys sikokianus* (Makino) Nakai, while *O. iwarenge* f. *magna* Y.N.Lee (Subsection *Appendiculata*) and three *Hylotelephium* H.Ohba species formed a group with strong support ([BS] = 100 and [PP] = 1.0).

## 4. Discussion

Ten new plastomes of *Crassula* were reported in the present study. Combined with available data from public database, we conducted comprehensive analyses, including plastome organizations, codon usage and aversion patterns, evolutionary rates, and phylogenetic implications.

The expansion and contraction of IR regions are common evolutionary events and have been considered as the main mechanism for the length variation of angiosperm plastomes [64,65,66]. In our study, we performed comparative analyses among *Crassula* plastomes, and found that the IRb regions had uniform length (110 bp) expansions to the *rps19* gene. This 110-bp expansion had been also observed in *Aeonium*, *Monanthes*, and most other taxa of Crassulaceae in our recent study [17]. This finding indicated that the conserved IR organization might act as a family-specific marker for Crassulaceae species.

Interestingly, it was reported that *rps19* genes were completely located in the LSC regions in *Forsythia suspensa* (Thunb.) Vahl, *Olea europaea* Hoffmanns. & Link L., and *Quercus litseoides* Dunn [67,68], and were fully encoded by the IR regions in *Polystachya adansoniae* Rchb.f., *Polystachya bennettiana* Rchb.f., and *Dracaena cinnabari* Balf.f. [69,70]. There are several mechanisms that might explain the IR expansion and contraction [71,72,73]. For instance, Goulding et al. [71] proposed that short IR expansions may occur by gene conversion events, whereas large IR expansions involved in double-strand DNA breaks. In order to better reveal the mechanisms of IR expansion and contraction, more extensive investigations in Crassulaceae and Saxifragales are required. 

Investigations of codon usage patterns could reveal phylogenetic relationships between organisms [25,74]. In particular, 11 species of *Crassula* can be divided into two subgenera from the RSCU heatmap, which agreed with the results of phylogenetic analyses. This finding further demonstrates that RSCU values contain phylogenetic implications [75,76,77,78,79,80]. Additionally, we observed codon usage patterns are gene-specific and/or species-specific, reflected in diversified ENC values and various distribution patterns in PR2 plots. Interestingly, we found the codon usage patterns of *matK* gene in *Crassula* species are unique among Crassulaceae species with elevated ENC values. Furthermore, the GC-biases of *matK* gene with specific preference (>0.5) might be the particular feature for subgenus *Crassula*. Due to rapid evolutionary rate, high universality, and significant interspecific divergence, the *matK* gene has been broadly used in plant evolutionary studies as one of the core DNA barcodes [9,10,81,82,83,84].

Codon aversion, a novel concept proposed by Miller et al. [29,30,31], is an informative character in phylogenetics. Specifically, the codon aversion motifs in orthologous genes are generally conserved in specific lineages [29,30,31]. To date, these analyses have only been performed in a few plant plastomes [17,26]. For example, the specific codon aversion motifs of *rpoA* gene could distinguish not only the two genera (*Aeonium* and *Monanthes*), but also the three subclades of *Aeonium* in our recent report [17]. In this work, genus-specific and subgenus-specific codon aversion motifs were identified for 11 *Crassula* species. These findings suggest codon aversion pattern could be used as a promising tool for phylogenetic study.

Generally, the dN/dS ratios of genes could reflect the extent of selection pressures during evolution [22]. Here, the dN/dS values of plastid PCGs ranged from 0 to 0.5891 within Crassulaceae, indicating all plastid genes were under purifying selection. Among these values, elevated dN/dS ratios were found for *ycf1* (0.4349) and *ycf2* (0.5891). Similarly, high dN/dS ratios of these two genes were also observed in other families, such as Asteraceae Bercht. & J.Presl [38], Mazaceae Reveal [22], and Musaceae Juss. [13]. The *ycf1* gene was related to protein translocation [85]. The *ycf2* gene is necessary for cell viability, but the detail function is still unknown [86]. Why *ycf1* and *ycf2* evolve relatively fast is interesting. The possible reason put forwarded by Barnard-Kubow et al. [87] considered that relaxed purifying selection or positive selection on *ycf1*, *ycf2* and some other genes might result in the development of reproductive isolation and subsequent speciation in plants. Therefore, the results suggested that *ycf1* and *ycf2* might play important roles in the divergence of Crassulaceae.

Our phylogenetic tree divided 87 species into 3 subfamilies and 7 clades. The clade Crassula is sister to all other 6 clades, which agrees with the phylogeny reported by Gontcharova et al. [4], Chang et al. [6], and Han et al. [17]. Furtherly, 11 *Crassula* species could be furtherly divided into two subgenera, which generally accords with the morphological differences (floral shape) reported by Bruyns et al. [10] (Appendix A). Nevertheless, there are still some unsolved phylogenetic problems within Crassulaceae. The first problem is that the plastid phylogeny of *Crassula* is not entirely clear due to the limited data. According to the classification proposed by Tölken [11,88], 11 and 9 sections were respectively identified in subgenus *Crassula* and subgenus *Disporocarpa*. However, Bruyns et al. [10] indicated that most sections were not monophyletic. Moreover, subgenus *Disporocarpa* recently has been regarded as a paraphyletic group [9,10]. The second is the genus *Sedum*, which is not monophyletic in our study, agreeing with the widely accepted viewpoint [3,4,5,89,90]. Finally, the genus *Orostachys* has been demonstrated to be non-monophyletic based on plastid data, which is consistent with previous analysis based on nuclear internal transcribed spacers (ITS) data [63]. In order to better understand the phylogeny of *Crassula* or Crassulaceae, more data are needed for the further detailed analyses.

## 5. Conclusions

In the present study, 10 new plastomes of *Crassula* species were reported. These plastomes exhibited identical gene content and order, and that they contained 134 genes (130 functional gene and 4 pseudogenes). The 11 identified HVRs with relatively high variability (π > 0.06886) might be used as potential DNA barcodes for species identification within *Crassula*. The unique expansion pattern, where the IRb regions had uniform length (110 bp) boundary expansions to *rps19*, might become a plesiomorphy of Crassulaceae. According to RSCU values, the A/T-ending codons were favored in plastid genes. Most importantly, we found the codon usage patterns of the *matK* gene in *Crassula* species are unique among Crassulaceae species with elevated ENC values. Furthermore, subgenus *Crassula* species have specific GC-biases in the *matK* gene. In addition, the codon aversion motifs from *matK*, *pafI* and *rpl22* contained phylogenetic implications within *Crassula*. Compared with other Crassulaceae species, 27 and 16 unique codon aversion motifs were detected for subgenus *Crassula* and subgenus *Disporocarpa*, respectively. Additionally, the evolutionary rates analyses indicated all plastid genes of Crassulaceae were under purifying selection. Among these genes, *ycf1* (dN/dS = 0.4349) and *ycf2* (dN/dS = 0.5891) were the most rapidly evolving genes, whereas *psaC* (dN/dS = 0) was the most conserved gene. Finally, our phylogenetic analyses strongly supported *Crassula* is sister to all other Crassulaceae species. Our results will be benefit for further evolutionary studies within the *Crassula* and Crassulaceae.

## Figures and Tables

**Figure 1 biology-11-01779-f001:**
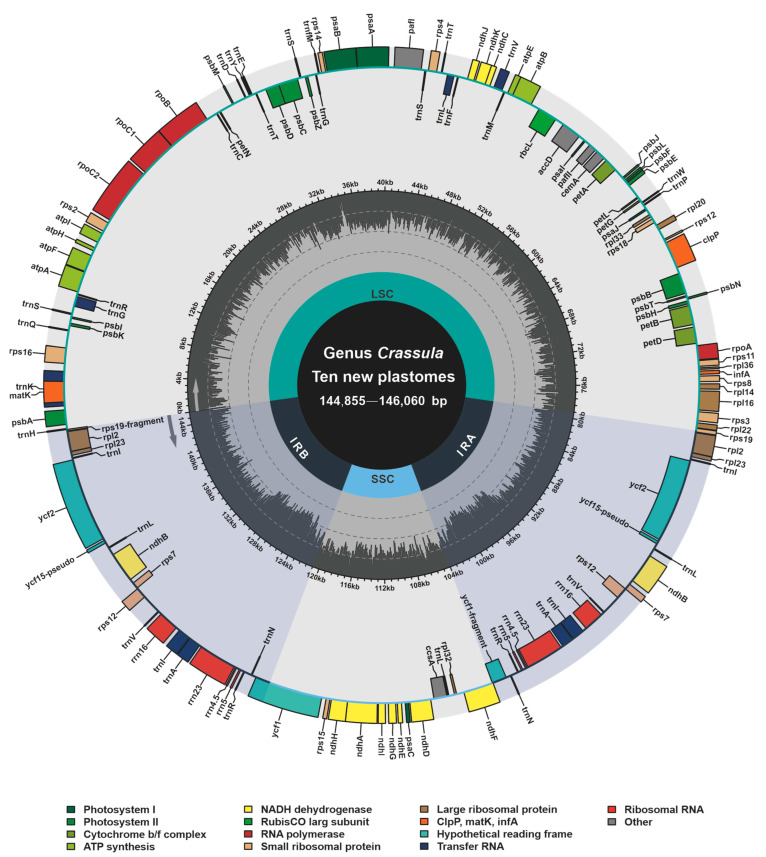
Annotation map of ten new plastomes from *Crassula* species. Directed with arrows, genes that are listed inside and outside of the circle are respectively transcribed clockwise and counterclockwise. Different colors represent different functional groups.

**Figure 2 biology-11-01779-f002:**
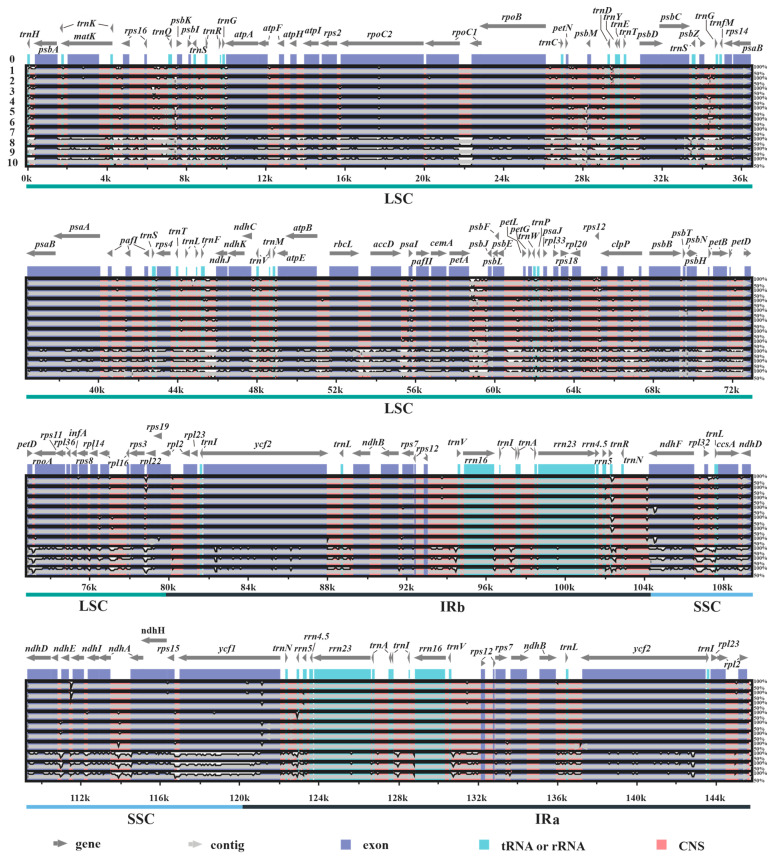
Structure comparisons of ten new *Crassula* plastomes using the mVISTA program. Y-scale represents the percent identity between 50% and 100%. The labels 0 to 10 indicate *C. perforata* (reference), *C. alstonii*, *C. columella*, *C. dejecta*, *C. mesembryanthoides*, *C. tecta*, *C. mesembrianthemopsis*, *C. socialis*, *C. volkensii*, *C. expansa* subsp. *fragilis*, and *C. deltoidei*, respectively.

**Figure 3 biology-11-01779-f003:**
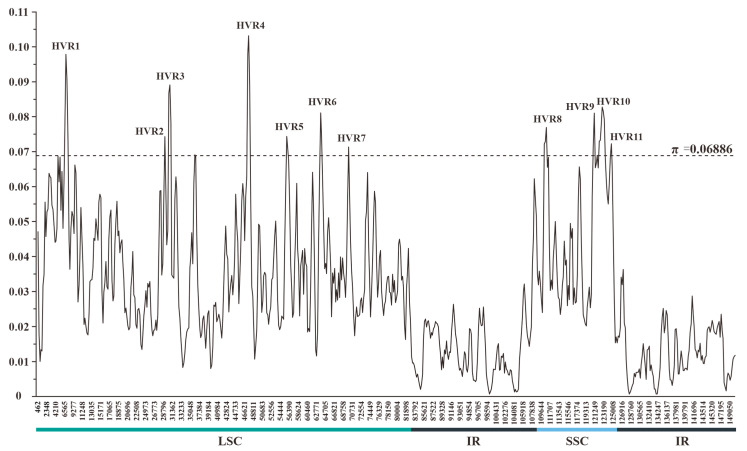
Sliding-window analysis of the plastomes of 11 *Crassula* species (window length: 600 bp; step size: 200 bp). x-axis: position of the midpoint of a window; y-axis: π value of each window. Regions with higher π values (π > 0.06886) were considered as HVRs.

**Figure 4 biology-11-01779-f004:**
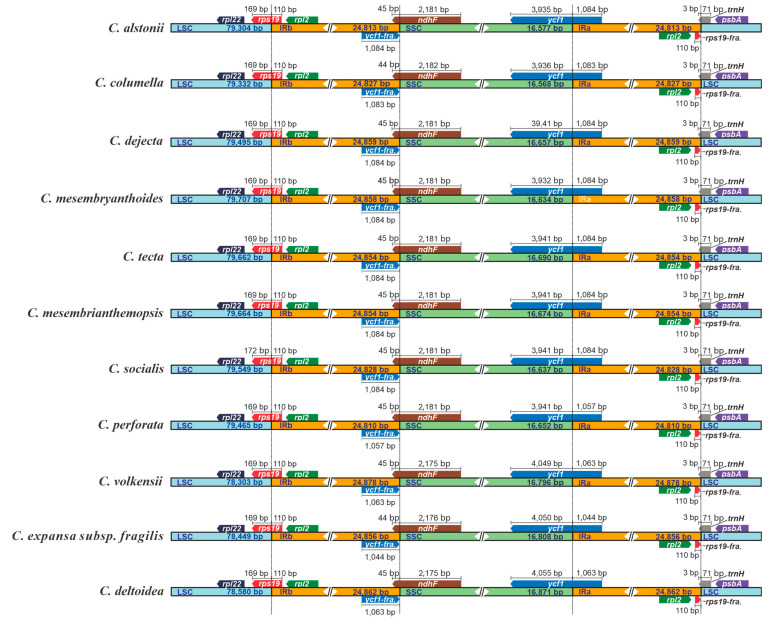
Comparisons of LSC, SSC, and IR region borders among plastomes of 11 *Crassula* species. Blue, orange and green blocks represent the LSC, IR and SSC regions, respectively. Gene boxes represented above the block were transcribed clockwise and those represented below the block were transcribed clockwise. “fra.” is the abbreviation of “fragment”.

**Figure 5 biology-11-01779-f005:**
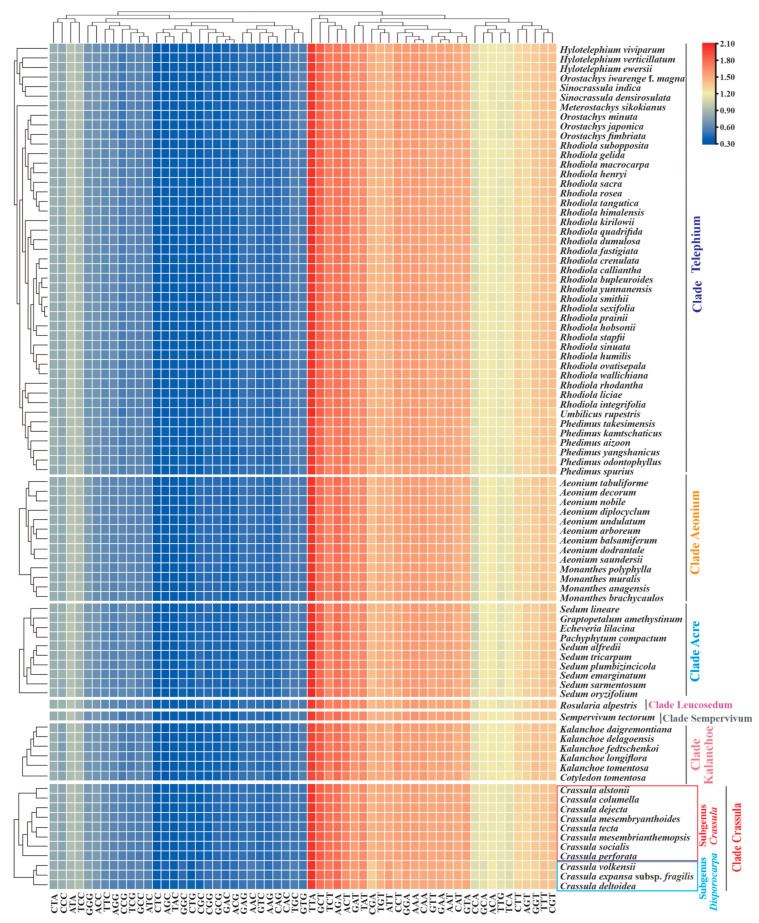
The heatmap of overall RSCU values among 7 clades of Crassulaceae species based on 53 concatenated plastid genes (length ≥300 bp). The x-axis: the cluster of different codons; y-axis: the clusters of species.

**Figure 6 biology-11-01779-f006:**
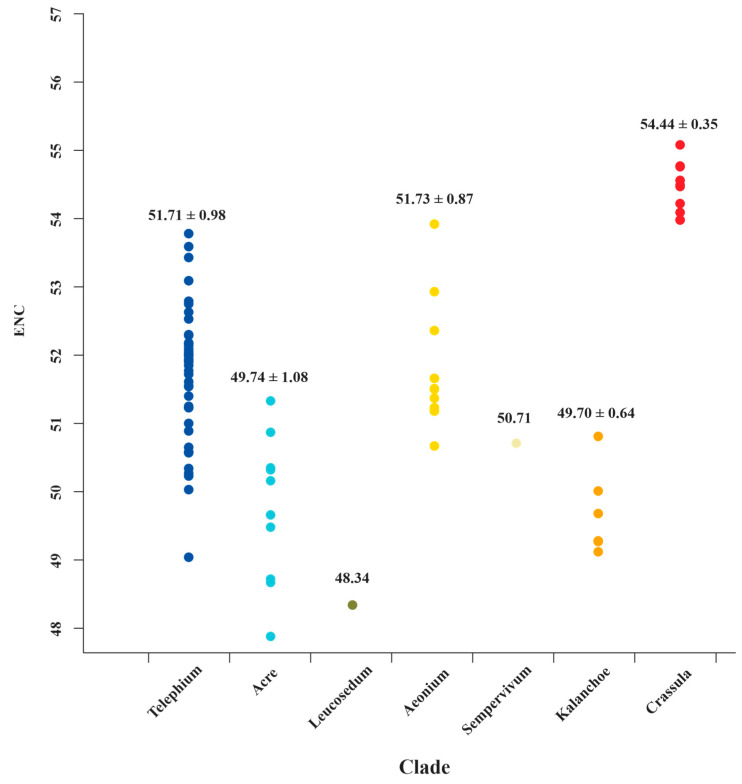
The ENC value distributions of *matK* for 7 clades of Crassulaceae. The mean values with standard deviations are labeled.

**Figure 7 biology-11-01779-f007:**
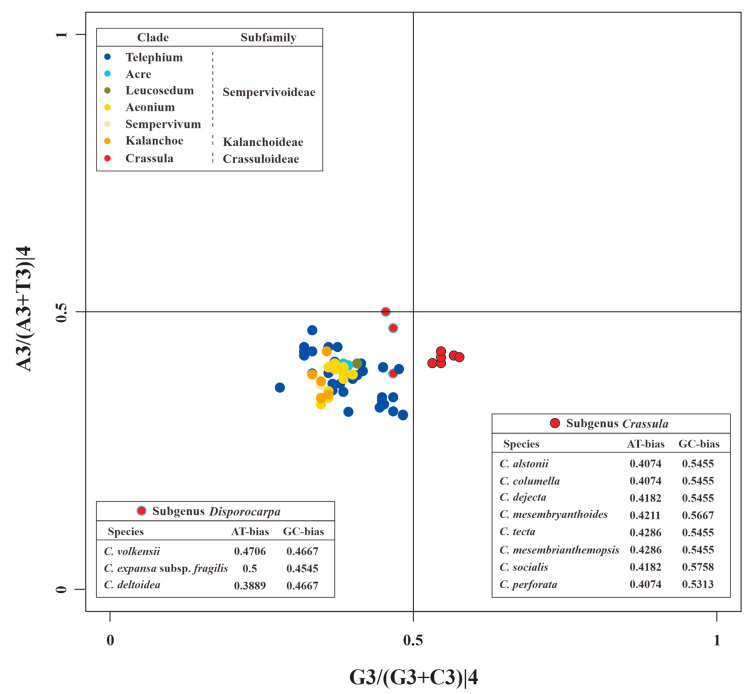
The PR2 plot of *matK* of Crassulaceae. Different colors represent different clades. Within the clade Crassula (or genus *Crassula*), red circles with black edges and cyan edges represent species of subgenus *Crassula* and *Disporocarpa*, respectively.

**Figure 8 biology-11-01779-f008:**
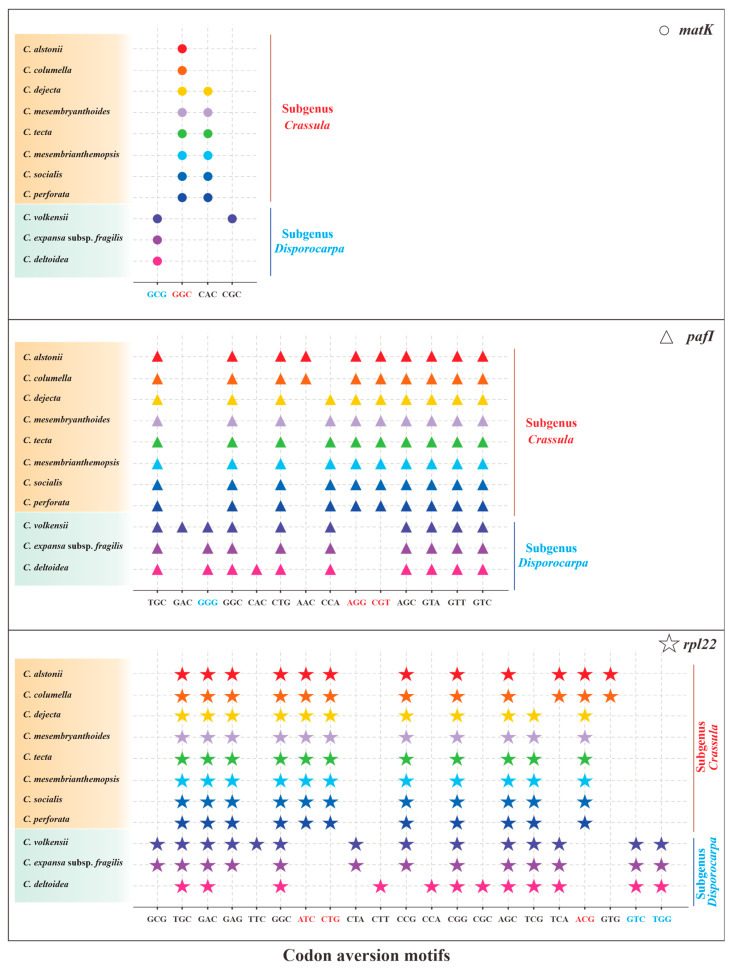
The specific codon aversion motifs of *matK*, *pafI* and *rpl22* gene for the 11 species of *Crassula*. The dots marked in different colors represent different species. Codons in red and green were specific for subgenus *Crassula* and subgenus *Disporocarpa*, respectively.

**Figure 9 biology-11-01779-f009:**
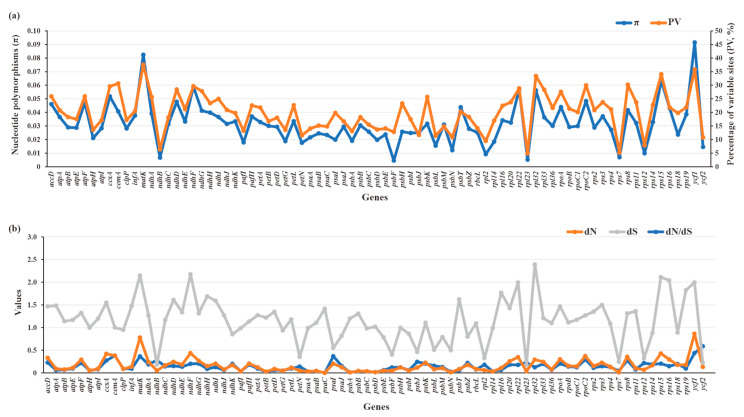
Sequence polymorphism among 79 PCGs of 87 Crassulaceae species. (**a**) Nucleotide diversity (π) and percentages of variable sites (PV). (**b**) Estimations of nonsynonymous (dN) and synonymous (dS) substitution rates and the dN/dS.

**Figure 10 biology-11-01779-f010:**
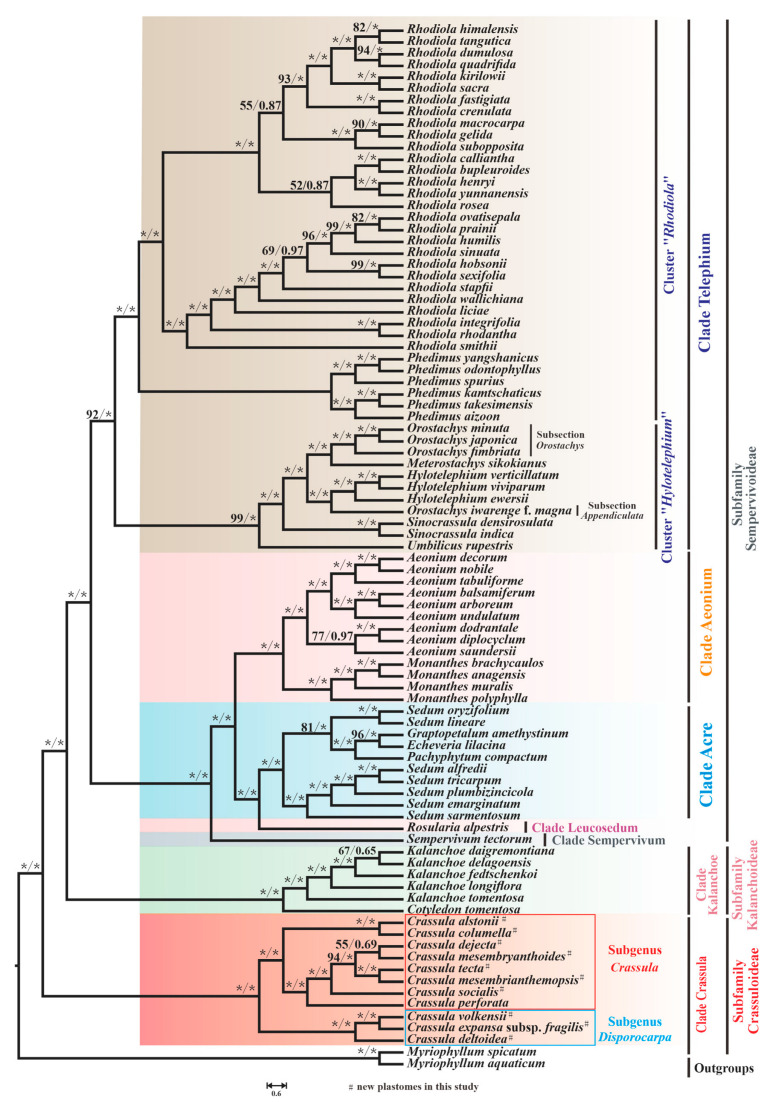
The phylogenetic tree of 87 Crassulaceae species based on ML and BI methods. The maximum likelihood bootstrap (BS) and Bayesian posterior probability (PP) values for each node are indicated; * indicates 100% bootstrap or 1.00 PP.

**Table 1 biology-11-01779-t001:** Plastome characteristics of *Crassula* species.

Species	Accession Number	Size (bp)				GC Content (%)				Gene Number				
		Genome	LSC	IR	SSC	Genome	LSC	IR	SSC	Total	PCGs	tRNA	rRNA	Pseudo
*C. alstonii* ^†^	OP729482 *	145,507	79,304	24,813	16,577	37.82	35.86	42.96	31.85	134 (19)	85 (6)	37 (7)	8 (4)	4 (2)
*C. columella* ^†^	OP729483 *	145,554	79,332	24,827	16,568	37.83	35.86	42.95	31.86	134 (19)	85 (6)	37 (7)	8 (4)	4 (2)
*C. dejecta* ^†^	OP729484 *	145,870	79,495	24,859	16,657	37.73	35.76	42.93	31.67	134 (19)	85 (6)	37 (7)	8 (4)	4 (2)
*C. mesembrianthemopsis* ^†^	OP882297 *	146,046	79,664	24,854	16,674	37.78	35.78	42.97	31.89	134 (19)	85 (6)	37 (7)	8 (4)	4 (2)
*C. mesembryanthoides* ^†^	OP729485 *	146,057	79,707	24,858	16,634	37.78	35.78	42.94	31.9	134 (19)	85 (6)	37 (7)	8 (4)	4 (2)
*C. perforata* ^†^	NC_053949	145,737	79,465	24,810	16,652	37.75	35.75	42.97	31.77	134 (19)	85 (6)	37 (7)	8 (4)	4 (2)
*C. socialis* ^†^	OP729486 *	145,842	79,549	24,828	16,637	37.79	35.8	42.95	31.86	134 (19)	85 (6)	37 (7)	8 (4)	4 (2)
*C. tecta* ^†^	OP729487 *	146,060	79,662	24,854	16,690	37.78	35.78	42.97	31.88	134 (19)	85 (6)	37 (7)	8 (4)	4 (2)
*C. deltoidea* ^#^	OP882298 *	145,175	78,580	24,862	16,871	38.13	36.25	43.08	32.32	134 (19)	85 (6)	37 (7)	8 (4)	4 (2)
*C. expansa subsp. Fragilis* ^#^	OP882299 *	144,969	78,449	24,856	16,808	38.30	36.51	43.14	32.4	134 (19)	85 (6)	37 (7)	8 (4)	4 (2)
*C. volkensii* ^#^	OP882300 *	144,855	78,303	24,878	16,796	38.32	36.51	43.15	32.4	134 (19)	85 (6)	37 (7)	8 (4)	4 (2)

^†^ These species belong to subgenus *Crassula*. ^#^, These speciesbelong to subgenus *Disporocarpa*. *, The new plastomes were generated in this study. (n), The number of genes located on IRs.

**Table 2 biology-11-01779-t002:** The HVRs identified in the plastomes of 11 *Crassula* species.

HVR	Coordinates	Region Size (bp)	π Value	Variable Sites	Annotations
HVR1	6259–8197	1939	0.07521	240	*rps16*–*trnQ-UUG*
HVR2	28,684–29,678	995	0.07427	119	*petN*–*psbM*
HVR3	29,942–31,262	1321	0.07064	182	*psbM*–*trnD-GUC*–*trnY-GUA*
HVR4	35,624–36,637	1014	0.06912	114	*psbZ*–*trnG-GCC*
HVR5	47,003–48,289	1287	0.08653	223	*trnL-UAA*–*trnF-GAA*–*ndhJ*
HVR6	55,345–56,290	946	0.07518	158	*rbcL*–*accD*
HVR7	63,273–64,156	884	0.07652	164	*psbE*–*petL*
HVR8	69,495–70,212	718	0.07133	111	*clpP*–*psbB*
HVR9	109,945–111,098	1154	0.06969	192	*ndhF*–*rpl32*–*trnL-UAG*
HVR10	120,441–123,490	3050	0.07259	559	*rps15*–*ycf1*
HVR11	124,303–124,908	606	0.07227	119	*ycf1*

**Table 3 biology-11-01779-t003:** The specific codon aversion motifs of *Crassula* within Crassulaceae.

Gene	Specific Codon Aversion Motifs		
	Subgenus *Crassula*	Subgenus *Disporocarpa*	Consensus for Genus *Crassula*
*accD*	TGC		
*atpB*	TGT	CGG	
*atpI*	TGT, CTG, CTA,		
*cemA*	GGG		
*ccsA*	GGC		
*ndhA*	CAT, CCG, CAG, CGG	CAT, CCG	CAT, CCG
*ndhE*	CTC		
*ndhI*	TCT	TCT	TCT
*ndhJ*	TGT, ACG		
*ndhK*	TGC, CAC, CTG	TGC, CAC, CTG	TGC, CAC, CTG
*pafII*	AAG	AAG	AAG
*petB*	AAC, AGA	CAG	
*petD*	GGC	GCC	
*psbB*		CGA	
*psbD*		ACG	
*rbcL*	GCG, CCC, AGG		
*rpl16*	GAT	GAT	GAT
*rpl20*		TGT	
*rps3*		GCG, GTC	
*rps4*	TCT	CGG	

## Data Availability

The sequence data generated in this study are available in GenBank of the National Center for Biotechnology Information (NCBI) under the access numbers: OP729482–OP729487 and OP882297–OP882300.

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
