# Peer review of "Ten Plastomes of Crassula (Crassulaceae) and Phylogenetic Implications"

_biology, 2022, doi:10.3390/biology11121779_

Round 1

Reviewer 1 Report

The authors proposed for publication the ms. "Comparative Plastomes Analyses of Crassula Species and Phylogenetic Relationships within Crassulaceae", aiming at comparing six complete plastid genomes (hitherto unpublished) with phylogenetic \ taxonomic and barcoding purposes.

The work was accurately planned, rigorously carried out and well written. The methods are state-of-the-art and the purposes are interesting, achieved and worth of publication on an international journal.

I recommend the acceptance of the ms after a thorough revision of nomenclature, and improvements in the M&M section (in order to facilitate other scholars) and especially in the Introduction and in the Discussion.

Unfortunately, due to some inaccuracy of the present version, the ms. risks to be unfairly judged as superficial by taxonomists.

First, the plant names must be accompanied, at least at their first occurrence, by the correct authorships.  The same for formal groups as genus, up to family. In second place, genus and combinations must be written in italics or an otherwise distinguishable form as compared to the other text. This is mandatory.

The introduction does not include relevant information about the taxonomy of Crassula. A brief historical background would be appreciable, albeit not indispensable; however, the most recent taxonomic results and treatments must be reported. 

As far as the discussion is involved: the two clades found in Crassula were already suggested by previous authors? Do these clades roughly correspond to taxonomic groups? What about the polyphyly of Sedum: do by any chance the species which fall "outside" Sedum all belong to a single taxonomic group? 

In addition, expound further how your results could help in identification of doubtful individual\taxa (barcoding).

Other comments:

1. Please, add further details about Bayesian Analysis (i.e., Burn-In) and state in the main text the best-fit evolutionary models included in S6.

2. Please, justify the choice of Halogaraceae as outgroup.

3. Please, replace the word "platome" with "plastome" in the Simple Summary.

Author Response

Dear Dr. Reviewer 1,

Thank you for your professional comments and suggestions to our manuscript entitled “Comparative Plastomes Analyses of Crassula Species and Phylogenetic Relationships within Crassulaceae” (#biology-2042165: major revisions). According to the suggestion proposed by Reviewer 2 and Reviewer 3, more new data have been added in this study. So, in order to describe our results more scrupulously, we have changed the title as “Ten Plastomes of Crassula (Crassulaceae) and Phylogenetic Implications” (new). We have modified the manuscript accordingly; Revised portion are marked in “Revised Manuscript with Track Changes” and detailed corrections are listed below point by point.

Response to Reviewer 1 Comments

Main point: First, the plant names must be accompanied, at least at their first occurrence, by the correct authorships. The same for formal groups as genus, up to family. In second place, genus and combinations must be written in italics or an otherwise distinguishable form as compared to the other text. This is mandatory.

The introduction does not include relevant information about the taxonomy of Crassula. A brief historical background would be appreciable, albeit not indispensable; however, the most recent taxonomic results and treatments must be reported.

As far as the discussion is involved: the two clades found in Crassula were already suggested by previous authors? Do these clades roughly correspond to taxonomic groups? What about the polyphyly of Sedum: do by any chance the species which fall "outside" Sedum all belong to a single taxonomic group?

In addition, expound further how your results could help in identification of doubtful individual\taxa (barcoding).

Response:

  1. Thanks for your good recommendation. We have added the authors for families, subfamilies, genera, subgenera, and species. The replacements were listed below:

Families:

“Orchidaceae” → “Orchidaceae Juss.”

“Crassulaceae” → “Crassulaceae J.St.-Hil”

“Asteraceae” → “Asteraceae Bercht. & J.Presl”

“Mazaceae” → “Mazaceae Reveal”

“Musaceae” → “Musaceae Juss.”

“Haloragaceae” → “Haloragaceae R.Br.”

Subfamilies:

“Crassuloideae” → “Crassuloideae Burnett”

“Kalanchoideae” → “Kalanchoideae A. Berger”

“Sempervivoideae” → “Sempervivoideae Arn. ”

Genera:

Crassula” → “Crassula L.”

Aeonium” → “Aeonium Webb & Berthel. ”

Monanthes” → “Monanthes Haw.”

Orostachys” → “Orostachys Fisch.”

Hylotelephium” Hylotelephium H.Ohba”

Sedum” → “Sedum L.”

Bletilla” → “Bletilla Rchb.f.”

Kalanchoe” Kalanchoe Adans.”

Cotyledon” Cotyledon L.”

Graptopetalum” Graptopetalum Rose”

Echeveria” Echeveria DC.”

Pachyphytum” Pachyphytum Link, Klotzsch & Otto”

Subgenera:

Crassula” → “Crassula L.”

Disporocarpa” → “Disporocarpa Fischer & C.A. Mey.”

Species:

C. alstonii” C. alstonii Marloth”

C. columella” C. columella Marloth & Schönland”

C. dejecta” C. dejecta Jacq.”

C. mesembryanthoides” C. mesembryanthoides (Haw.) D.Dietr.”

C. socialis” C. socialis Schönland”

C. tecta” C. tecta Thunb.”

C. perforate” C. perforata Thunb.”

Myriophyllum aquaticum” Myriophyllum aquaticum (Vell.) Verdc.”

Myriophyllum spicatum” Myriophyllum spicatum L.”

Sedum sarmentosum” Sedum sarmentosum Bunge”

O. japonica” O. japonica (Maxim.) A.Berger”

O. minuta” O. minuta (Kom.) A.Berger”

O. fimbriata” O. fimbriata (Turcz.) A.Berger”

Meterostachys sikokianus” Meterostachys sikokianus (Makino) Nakai”

O. iwarenge” O. iwarenge f. magna Y.N.Lee”

Forsythia suspensa” Forsythia suspensa (Thunb.) Vahl”

Olea europaea” Olea europaea Hoffmanns. & Link L.”

Quercus litseoides” Quercus litseoides Dunn”

Polystachya adansoniae” Polystachya adansoniae Rchb.f.”

Polystachya bennettiana” Polystachya bennettiana Rchb.f.”

Dracaena cinnabari” Dracaena cinnabari Balf.f.”

  1. The reviewer raised a professional and valuable suggestion. The brief historical background of the taxonomy of Crassula was added in the introduction part. The revised part was as following:

“Previous taxonomic revision of genus Crassula recognized two subgenera: Crassula L. and Disporocarpa Fischer & C.A. Mey. [7,11,12]. The monophyly of the subgenus Crassula was well supported in two recent molecular phylogenetic studies [9,10]. Nevertheless, the monophyly of subgenus Disporocarpa is still controversial [9,10]. Thus, more evidences and further investigations are required to clarify the phylogenetic relationships of genus Crassula.”

  1. Thanks for your professional suggestion. According to the recommendation presented by Reviewer 2 and Reviewer 3, we have added four new data for this study. Thus, we could cluster 11 Crassula species (10 are new in our study) into two subgenera: subgenus Crassula and subgenus Disporocarpa. The current two subgenera correspond to the accepted taxonomic groups. In addition, the Sedum, containing approximately 470 species, is by far the largest genus of Crassulaceae. However, in current circumscription, Sedum indeed is a highly polyphyletic taxon. Recently, Messerschmid et al. (10.1002/tax.12316) considered that the taxonomy of Sedum might be “Linnaeus’s folly”. As mentioned above, the family Crassulaceae could be divided into seven clades. Notably, except for clade Crassula and Kalanchoe, all the remaining five clades contained species of Sedum. Fortunately, these inaccurate taxa have been revised step by step. For example, species of Phedimus, previously treated as members of Sedum, have been classified as a separate genus. So, species which fall "outside" Sedum might belong to a single taxonomic group. For further taxonomic revision of Sedum, more data are needed in future studies.

  1. As your professional suggestion as well as the recommendation presented by Reviewer 2, we have added the highly variable region selection for potential DNA barcoding. The revised part in our manuscript was as following:

Figure 3. Sliding window analysis of the plastomes of 11 species from genus Crassula (window length: 600 bp; step size: 200 bp). x-axis: position of the midpoint of a window; y-axis: π value of each window. Regions with higher π values (π > 0.06886) were considered as HVRs.

Table 2. The HVRs identified in the plastomes of 11 species from genus Crassula.

HVR

Coordinates

Region size (bp)

π value

Variable sites

Annotations

HVR1

6259−8197

1939

0.07521

240

rps16trnQ-UUG

HVR2

28,684−29,678

995

0.07427

119

petNpsbM

HVR3

29,942−31,262

1321

0.07064

182

psbMtrnD-GUCtrnY-GUA

HVR4

35,624−36,637

1014

0.06912

114

psbZtrnG-GCC

HVR5

47,003−48,289

1287

0.08653

223

trnL-UAAtrnF-GAAndhJ

HVR6

55,345−56,290

946

0.07518

158

rbcLaccD

HVR7

63,273−64,156

884

0.07652

164

psbEpetL

HVR8

69,495−70,212

718

0.07133

111

clpPpsbB

HVR9

109,945−111,098

1154

0.06969

192

ndhFrpl32trnL-UAG

HVR10

120,441−123,490

3050

0.07259

559

rps15ycf1

HVR11

124,303−124,908

606

0.07227

119

ycf1

“The sliding window-based π values estimated by 11 plastomes of genus Crassula ranged from 0.00073 to 0.10315 (Table S2 and Data S2). The mean π value and its standard deviation were 0.02978 and 0.01954, respectively. Thus, a total of 11 HVRs were identified with relatively high variability (π > 0.06886) (Figure 3). These HVRs containing high π values (0.06912−0.08653) and abundant variable sites (111−559) might be used as potential DNA barcodes for species identification within genus Crassula (Table 2).”

Other comments:

  1. Please, add further details about Bayesian Analysis (i.e., Burn-In) and state in the main text the best-fit evolutionary models included in S6.

Response: Thanks for your professional and valuable suggestions. We have added the details about Bayesian Analysis and best-fit evolutionary models. The revised part in our manuscript was as following:

In methods part:

“After discarding the first 25% trees as burn-in, the remaining 75% trees were used to estimate the consensus tree and Bayesian posterior probabilities.”

In results part:

“After model test, GTR+G4 and GTR+I+G4 were inferred as the optimal substitution models for most genes (the detailed models could be seen in Table S7).”

  1. Please, justify the choice of Halogaraceae as outgroup.

Response: Thanks for your kind recommendation. According to the studies of Bruyns et al. (10.1016/j.ympev.2018.10.045) and Lu et al. (10.1002/ajb2.1797), the family the Haloragaceae is the sister-family of Crassulaceae. Therefore, we selected species from Haloragaceae as outgroups. The revised part in our manuscript was as following:

“Recent studies of Lu et al. [9] and Bruyns et al. [10] revealed a sister relationship between Crassulaceae and Haloragaceae R.Br.. Therefore, two species of Haloragaceae (Myriophyllum aquaticum (Vell.) Verdc., NC_048889; and Myriophyllum spicatum L., NC_037885) were selected as outgroups.”

  1. Please, replace the word "platome" with "plastome" in the Simple Summary.

Response: We accepted this comment sincerely. We have revised this mistake.

Reviewer 2 Report

This study newly sequenced six plastomes of Crassula. The comparative analyses showed unique codon usage and aversion patterns of Crassula plastomes. Evolutionary rates and phylogenetic relationships of Crassula were also analyzed using plastid sequences. The findings revealed structural characterizations and evolution history of Crassula plastomes, and provide potential molecular markers for DNA barcoding. However, there were several concerns about this article before the manuscript can be published in Biology.

 Major concerns:

1. More newly sequenced plastomes were probably needed for phylogenetic analyses of Crassula. 

2. In the keywords, DNA barcoding mentioned does not seem to be described too much in the article. Please add the analysis of highly variable region selection and DNA barcoding screening. 

Minor concerns:

Line 45-53: It would be better to describe whether the classification of Crassulaceae is difficult or controversial, especially in Crassula. It is an important reason why you did this study. 

line 73: The abbreviation of the “CAM” probably is ambiguous, because “CAM” is wildly considered as crassulacean acid metabolism. Therefore, using the full name would be better. 

Line 111: The parameters of the software need to be provided. 

Line 111-117: The nodes between LSC/SSC/IR needed experimental verification. 

Figure 1-7: Detailed legends of figures needed to be provided. 

Figure 9: The number of Crassulaceae species included in phylogenetic analyses is necessary for a complete legend of the figure. 

Reference: Journal names should be kept uniform, abbreviated or full.

Author Response

Dear Dr. Reviewer 2,

Thank you for your professional comments and suggestions to our manuscript entitled “Comparative Plastomes Analyses of Crassula Species and Phylogenetic Relationships within Crassulaceae” (#biology-2042165: major revisions). We have modified the manuscript accordingly; Revised portion are marked in “Revised Manuscript with Track Changes” and detailed corrections are listed below point by point.

Response to Reviewer 2 Comments

Major concerns:

  1. More newly sequenced plastomes were probably needed for phylogenetic analyses of Crassula.

Response: Thanks for your kind recommendation. We have added four new plastomes of genus Crassula in the revised manuscript. The four new plastomes with their accession numbers were as following:

Subgenus Crassula:

Crassula mesembrianthemopsis (OP882297)

Subgenus Disporocarpa:

Crassula deltoidei (OP882298)

Crassula expansa subsp. fragilis (OP882299)

Crassula volkensii (OP882300)

In order to describe our results more scrupulously, we have changed the title from “Comparative Plastomes Analyses of Crassula Species and Phylogenetic Relationships within Crassulaceae” to “Ten Plastomes of Crassula (Crassulaceae) and Phylogenetic Implications”. As your good suggestion, we would like to further collect more samples in genus Crassula.

  1. In the keywords, DNA barcoding mentioned does not seem to be described too much in the article. Please add the analysis of highly variable region selection and DNA barcoding screening.

Response: Thanks for your professional and valuable suggestion. We highly agree with your viewpoint. Crassula is a species rich and morphologically diverse genus. Therefore, accurate species delimitation at the molecular level is essential for this taxon.

According to your good suggestion, we have added the highly variable region (HVR) analysis for plastomes of genus Crassula. The detail data could be seen in Table S2 and Data S2. The new revised part in our manuscript was as following:

Figure 3. Sliding window analysis of the plastomes of 11 species from genus Crassula (window length: 600 bp; step size: 200 bp). x-axis: position of the midpoint of a window; y-axis: π value of each window. Regions with higher π values (π > 0.06886) were considered as HVRs.

Table 2. The HVRs identified in the plastomes of 11 species from genus Crassula.

HVR

Coordinates

Region size (bp)

π value

Variable sites

Annotations

HVR1

6259−8197

1939

0.07521

240

rps16trnQ-UUG

HVR2

28,684−29,678

995

0.07427

119

petNpsbM

HVR3

29,942−31,262

1321

0.07064

182

psbMtrnD-GUCtrnY-GUA

HVR4

35,624−36,637

1014

0.06912

114

psbZtrnG-GCC

HVR5

47,003−48,289

1287

0.08653

223

trnL-UAAtrnF-GAAndhJ

HVR6

55,345−56,290

946

0.07518

158

rbcLaccD

HVR7

63,273−64,156

884

0.07652

164

psbEpetL

HVR8

69,495−70,212

718

0.07133

111

clpPpsbB

HVR9

109,945−111,098

1154

0.06969

192

ndhFrpl32trnL-UAG

HVR10

120,441−123,490

3050

0.07259

559

rps15ycf1

HVR11

124,303−124,908

606

0.07227

119

ycf1

“The sliding window-based π values estimated by 11 plastomes of genus Crassula ranged from 0.00073 to 0.10315 (Table S2 and Data S2). The mean π value and its standard deviation were 0.02978 and 0.01954, respectively. Thus, a total of 11 HVRs were identified with relatively high variability (π > 0.06886) (Figure 3). These HVRs containing high π values (0.06912−0.08653) and abundant variable sites (111−559) might be used as potential DNA barcodes for species identification within genus Crassula (Table 2).”

Minor concerns:

  1. Line 45-53: It would be better to describe whether the classification of Crassulaceae is difficult or controversial, especially in Crassula. It is an important reason why you did this study.

Response: The reviewer raised a professional and valuable suggestion. As your recommendation, the historical background of the taxonomy of Crassula was added in the introduction part. The revised part was as following:

“Previous taxonomic revision of genus Crassula recognized two subgenera: Crassula L. and Disporocarpa Fischer & C.A. Mey. [7,11,12]. The monophyly of the subgenus Crassula was well supported in two recent molecular phylogenetic studies [9,10]. Nevertheless, the monophyly of subgenus Disporocarpa is still controversial [9,10]. Thus, more evidences and further investigations are required to clarify the phylogenetic relationships of genus Crassula.”

  1. line 73: The abbreviation of the “CAM” probably is ambiguous, because “CAM” is wildly considered as crassulacean acid metabolism. Therefore, using the full name would be better.

Response: Many thanks for this comment. We agree with your viewpoint that “CAM” is wildly considered as crassulacean acid metabolism. Therefore, we have revised the words from “CAM” to “codon aversion motif”.

  1. The parameters of the software need to be provided.

Response: Thanks for your good recommendation. We have added more detailed parameters in this part.

  1. Line 111-117: The nodes between LSC/SSC/IR needed experimental verification.

Response: Thank you again for your comment. In angiosperms, the plastome generally exhibits a conserved quadripartite circular structure comprising of two single copy regions (LSC and SSC) and two inverted repeat regions (IRs). Similarly, the plastomes of Crassulaceae are also conservative as quadripartite structure molecules. For examples, all plastomes of Sedum plumbizincicola (clade Acre) (Ding et al. 2019) and 13 Macaronesian species (clade Aeonium) (Han et al. 2022) we reported were formed with four parts (LSC, SSC, IRa and IRb). Most importantly, similar IR junction patterns were observed in these taxa (Han et al. 2022).

In addition, GetOrganelle, one of the most successful tools, can de novo assemble the organelle genomes accurately (Jin et al. 2020). Recently, this tool had been recommended as the default option for plastome assemblies (Freudenthal et al. 2020). GeSeq is a fast web application that generates high-quality annotations for organelle genomes (Tillich et al. 2017). Therefore, we believed our assemblies and annotations of ten plastomes are reliable.

Moreover, as reported by our recent academic paper (Ding et al. 2022), larger inverted repeats might result in sequence redistribution (‘flip-flop’). So, two conformations of plastome could be seen in assembly results yielded by GetOrganelle. In order to analyse plastome conveniently, one conformation was uniformly selected as representative of plastome. It is an unwritten rule. In order to confirm the nodes between LSC/SSC/IR, long sequence reads generated by ONT or PacBio sequencing technologies are needed. In order to better understand the conformations of plastomes, we will sequence and analyse more plastomes by using ONT or PacBio sequencing platform. Thank you again for this comment.

References:

  1. Ding, H.; Zhu, R.; Dong, J.; Bi, D.; Jiang, L.; Zeng, J.; Huang, Q.; Liu, H.; Xu, W.; Wu, L. Next-Generation Genome Sequencing of Sedum plumbizincicola Sheds Light on the Structural Evolution of Plastid rRNA Operon and Phylogenetic Implications within Saxifragales. Plants 2019, 8, 386.
  2. Han, S.; Bi, D.; Yi, R.; Ding, H.; Wu, L.; Kan, X. Plastome evolution of Aeonium and Monanthes (Crassulaceae): Insights into the variation of plastomic tRNAs, and the patterns of codon usage and aversion. Planta 2022, 256, 35.
  3. Jin, J.-J.; Yu, W.-B.; Yang, J.-B.; Song, Y.; DePamphilis, C.W.; Yi, T.-S.; Li, D.-Z. GetOrganelle: a fast and versatile toolkit for accurate de novo assembly of organelle genomes. Genome biology 2020, 21, 241.
  4. Freudenthal, J.A.; Pfaff, S.; Terhoeven, N.; Korte, A.; Ankenbrand, M.J.; Förster, F. A systematic comparison of chloroplast genome assembly tools. Genome biology 2020, 21, 254.
  5. Tillich, M.; Lehwark, P.; Pellizzer, T.; Ulbricht-Jones, E.S.; Fischer, A.; Bock, R.; Greiner, S. GeSeq–versatile and accurate annotation of organelle genomes. Nucleic acids research 2017, 45, W6-W11.
  6. Ding, H.; Bi, D.; Zhang, S.; Han, S.; Ye, Y.; Yi, R.; Yang, J.; Liu, B.; Wu, L.; Zhuo, R. The Mitogenome of Sedum plumbizincicola (Crassulaceae): Insights into RNA Editing, Lateral Gene Transfer, and Phylogenetic Implications. Biology 2022, 11, 1661.

  1. Figure 1-7: Detailed legends of figures needed to be provided.

Response: We accepted this comment sincerely. The detailed figure legends were revised in the manuscript.

  1. Figure 9: The number of Crassulaceae species included in phylogenetic analyses is necessary for a complete legend of the figure.

Response: Thank you for your patient help. “87 Crassulaceae species” were added in the figure legend.

  1. Reference: Journal names should be kept uniform, abbreviated or full.

Response: Thank you again for your patient help. We have checked the references.

Reviewer 3 Report

The comparative plastomes analysis in this manuscript is very elegant, and the figures are very well-crafted. However, I have several questions: 1. The author did not collect samples from China, where some Crassula species are found, but instead focused on those found in Africa. There are several questions in this. First, whether the species identification is accurate;  Second, Where are these species which planted in the greenhouse come form? Such analyses of chloroplast structure in species of unknown origins provide no phylogenetic insights. 2. There are more than 200 species in Crassula, and the author sampled only 6 species which is far not enough. As the author said further research is needed. I suggest adding more materials to make the data representative. 3. The six species of Crassula were found to be divided into two groups by the phylogenetic tree. It is important for the authors to clarify the differences between the two groups in detail, such as morphological or geographical. 4. Taxonomically, the relationship between Crassula and Tillaea is unresolved, and further study is needed to clarify their relationship.

Author Response

Dear Dr. Reviewer 3,

Thank you for your professional comments and suggestions to our manuscript entitled “Comparative Plastomes Analyses of Crassula Species and Phylogenetic Relationships within Crassulaceae” (#biology-2042165: major revisions). We have modified the manuscript accordingly; Revised portion are marked in “Revised Manuscript with Track Changes” and detailed corrections are listed below point by point.

Response to Reviewer 3 Comments:

  1. The author did not collect samples from China, where some Crassula species are found, but instead focused on those found in Africa. There are several questions in this. First, whether the species identification is accurate; Second, where are these species which planted in the greenhouse come from? Such analyses of chloroplast structure in species of unknown origins provide no phylogenetic insights.

Response:

(1) The reviewer raised a professional and valuable suggestion. According to the “Flora of China”, four Tillaea species were distributed in China. By comparison with “Illustrated Handbook of Succulent Plants: Crassulaceae”, the accepted species names of these species were revised:

Tillaea aquatica and Tillaea yunnanensisCrassula aquatica

Tillaea mongolica Crassula decumbens var. decumbens

Tillaea pentandraCrassula laneeolata ssp. Dentieulata

We sincerely thank you for this comment. We had a strong desire to sample these species and would like to sequence the data of these Chinese native species in our future study.

(2) The current species we sampled were collected from professional organization, such as Koehres-Kakteen Online Shop (Erzhausen, Germany), Shanghai Chenshan Botanical Garden (Shanghai, China), Fairy Lake Botanical Garden (Shenzhen, China), and other Botanical Gardens of China. All species were planted in the greenhouse of Anhui Normal University. In addition, in order to accurately identify these species, molecular identifications were performed in advance by using phylogenetic method (ML method) based on 5 loci (ETS, ITS, rps16, matK, and trnL-F). The molecular identification results of species (including 4 new data) were as follow:

Crassula alstonii (ETS + ITS + rps16 + matK + trnL-F)

Crassula columella (ETS + ITS + rps16 + matK + trnL-F)

Crassula dejecta (ETS + ITS + rps16 + matK + trnL-F)

Note: According to the wikipedia, the closest species of C. dejecta is C. rubricaulis (https://en.wikipedia.org/wiki/Crassula_rubricaulis). Therefore, we considered our species identification is reliable.

Crassula deltoidea (ITS)

Crassula mesembryanthoides (ETS + ITS + rps16 + matK + trnL-F)

Crassula socialis (ITS)

Crassula tecta and Crassula mesembrianthemopsis (ETS + ITS + rps16 + matK + trnL-F)

Crassula volkensii and Crassula expansa subsp. fragilis (ETS + ITS + rps16 + matK + trnL-F)

To sum up, we believed our results are credible.

  1. There are more than 200 species in Crassula, and the author sampled only 6 species which is far not enough. As the author said further research is needed. I suggest adding more materials to make the data representative.

Response: Thanks for your kind recommendation. We have added four new plastomes of genus Crassula in the revised manuscript. The four new plastomes with their accession numbers were as following:

Subgenus Crassula:

Crassula mesembrianthemopsis (OP882297)

Subgenus Disporocarpa:

Crassula deltoidei (OP882298)

Crassula expansa subsp. fragilis (OP882299)

Crassula volkensii (OP882300)

In order to describe our results more scrupulously, we have changed the title from “Comparative Plastomes Analyses of Crassula Species and Phylogenetic Relationships within Crassulaceae” to “Ten Plastomes of Crassula (Crassulaceae) and Phylogenetic Implications”. As your good suggestion, we will further collect more samples in genus Crassula.

  1. The six species of Crassula were found to be divided into two groups by the phylogenetic tree. It is important for the authors to clarify the differences between the two groups in detail, such as morphological or geographical.

Response: Thanks for your good recommendation. Currently, new data were added in this study. Therefore, in our revised manuscript, ten new species we determined and Crassula perforate could be divided into two subgenera (Crassula and Disporocarpa). The obvious morphological differences between these two subgenera were reported, especially in floral shape. The floral shapes of subgenus Crassula species and subgenus Disporocarpa are urceolate to tubular and funnel- or cup-shaped, respectively. We have compared our molecule phylogenetic results with the morphological characteristics reported by Bruyns et al. (10.1016/j.ympev.2018.10.045). The detailed information was listed in Table S8. The revised part in discussion was as following:

“Furtherly, 11 genus Crassula species could be furtherly divided into two subgenera, which generally accords with the morphological differences (floral shape) reported by Bruyns et al. (Table S8).”

  1. Taxonomically, the relationship between Crassula and Tillaea is unresolved, and further study is needed to clarify their relationship.

Response: Thanks for your good recommendation. Recently, the relationship between Crassula and Tillaea is generally clear. To be specific, Tillaea now has been revised as Crassula. For example, Bywater and Wickens in 1984 indicated that Tillaea was treated as a synonym of Crassula. Similarly, the authoritative book “Illustrated Handbook of Succulent Plants: Crassulaceae” also treated Tillaea as a synonym of Crassula. Based on the molecular data, Mort et al. in 2009 indicated that Tillaea is neither monophyletic nor sister to Crassula. So, they also placed Tillaea within Crassula. In addition, all recent studies manifested that the Crassula is only unique genus in the clade Crassula (or subfamily Crassuloideae).

References:

  1. Bywater, M.; Wickens, G. New World species of the genus Crassula. Kew bulletin 1984, 699-728.
  2. Eggli, U. Crassulaceae. In Illustrated Handbook of Succulent Plants: Crassulaceae; Eggli, U., Ed.; Springer: Berlin/Heidelberg, Germany, 2003.
  3. Mort, M.E.; Randle, C.P.; Burgoyne, P.; Smith, G.; Jaarsveld, E.; Hopper, S.D. Analyses of cpDNA matK sequence data place Tillaea (Crassulaceae) within Crassula. Plant systematics and evolution 2009, 283, 211-217.

Thank you again for your suggestion, we will sample some species that have previously been included in the genus Tillaea.

Round 2

Reviewer 2 Report

This study newly sequenced six plastomes of Crassula. The comparative analyses showed unique codon usage and aversion patterns of Crassula plastomes. Evolutionary rates and phylogenetic relationships of Crassula were also analyzed using plastid sequences. The findings revealed structural characterizations and evolution history of Crassula plastomes, and provide potential molecular markers for DNA barcoding. The findings are interesting. The author has carefully answered the comments of the three reviewers. Now the manuscript has been greatly improved.

Author Response

Response to Reviewer 2 Comments

Dear Dr. Reviewer 2,

Thank you for your professional comments to our manuscript entitled “Ten Plastomes of Crassula (Crassulaceae) and Phylogenetic Implications” (#biology-2042165: minor revisions). We have modified the manuscript accordingly; Revised portion are marked in “Revised Manuscript with Track Changes” and detailed corrections are listed below point by point.

Response to Reviewer 2 Comments

This study newly sequenced six plastomes of Crassula. The comparative analyses showed unique codon usage and aversion patterns of Crassula plastomes. Evolutionary rates and phylogenetic relationships of Crassula were also analyzed using plastid sequences. The findings revealed structural characterizations and evolution history of Crassula plastomes, and provide potential molecular markers for DNA barcoding. The findings are interesting. The author has carefully answered the comments of the three reviewers. Now the manuscript has been greatly improved.

Response: Thank you again for your comments. We have learnt more from your suggestions. According to your professional recommendations, we have improved the quality of our manuscript.

Reviewer 3 Report

The author added plastid genome data of four species. However, subg. Disporocarpa with nine sections is a paraphyletic group and subg. Crassula with eleven sections. So, the current materials are insufficient to discuss these issues. Therefore, the phylogenetic implications of this study should be approached with caution. In addition, in the revised version, the words "genus" and "family" should be deleted before Crassula and Crassulaceae.

Author Response

Response to Reviewer 3 Comments

Dear Dr. Reviewer 3,

Thank you for your professional comments and suggestions to our manuscript entitled “Ten Plastomes of Crassula (Crassulaceae) and Phylogenetic Implications” (#biology-2042165: minor revisions). We have modified the manuscript accordingly; Revised portion are marked in “Revised Manuscript with Track Changes” and detailed corrections are listed below point by point.

Response to Reviewer 3 Comments:

The author added plastid genome data of four species. However, subg. Disporocarpa with nine sections is a paraphyletic group and subg. Crassula with eleven sections. So, the current materials are insufficient to discuss these issues. Therefore, the phylogenetic implications of this study should be approached with caution. In addition, in the revised version, the words "genus" and "family" should be deleted before Crassula and Crassulaceae.

Response: The reviewer raised professional and valuable suggestions. We highly agree with your viewpoint. Thus, the phylogenetic implications of Crassula have been discussed more rigorously in the revised manuscripts. The revised part in discussion was as following:

“The first problem is that the plastid phylogeny of Crassula is not entirely clear due to the limited data. According to the classification proposed by Tölken [11,88], 11 and 9 sections were respectively identified in subgenus Crassula and subgenus Disporocarpa. However, Bruyns et al. [10] indicated that most sections were not monophyletic. Moreover, subgenus Disporocarpa recently has been regarded as a paraphyletic group [9,10].”

In addition, the unnecessary words "genus" and "family" have been deleted before Crassula and Crassulaceae, as your good recommendation.

Thank you again for your comments.
